# Nutrition and Hydration for High-Altitude Alpinism: A Narrative Review

**DOI:** 10.3390/ijerph20043186

**Published:** 2023-02-11

**Authors:** Ginés Viscor, Jordi Corominas, Anna Carceller

**Affiliations:** 1Secció de Fisiologia, Departament de Biologia Cel·Lular, Fisiologia i Immunologia, Facultat de Biologia, Universitat de Barcelona, 08028 Barcelona, Spain; 2International Federation of Mountain Guide Associations (UIAGM/IFMGA), CH-1920 Bern, Switzerland

**Keywords:** sports physiology, mountaineering, high-altitude, metabolism, nutrition, hydration

## Abstract

This report aims to summarise the scientific knowledge around hydration, nutrition, and metabolism at high altitudes and to transfer it into the practical context of extreme altitude alpinism, which, as far as we know, has never been considered before in the literature. Maintaining energy balance during alpine expeditions is difficult for several reasons and requires a deep understanding of human physiology and the biological basis for altitude acclimation. However, in these harsh conditions it is difficult to reconcile our current scientific knowledge in sports nutrition or even for mountaineering to high-altitude alpinism: extreme hypoxia, cold, and the logistical difficulties intrinsic to these kinds of expeditions are not considered in the current literature. Requirements for the different stages of an expedition vary dramatically with increasing altitude, so recommendations must differentiate whether the alpinist is at base camp, at high-altitude camps, or attempting the summit. This paper highlights nutritional recommendations regarding prioritising carbohydrates as a source of energy and trying to maintain a protein balance with a practical contextualisation in the extreme altitude environment in the different stages of an alpine expedition. More research is needed regarding specific macro and micronutrient requirements as well as the adequacy of nutritional supplementations at high altitudes.

## 1. Introduction

Alpinism is a challenging sport that takes place in complex environments that include low temperatures and humidity and high altitudes, in addition to solar and cosmic radiation added to the exposure to objective dangers such as rock falls or avalanches. Apart from the physical requirements, which are the result of a mixture of disciplines such as rock climbing, ice climbing, ski mountaineering, and glacier travelling, the success highly depends on managing the risk exposure and decision making [1].

High-altitude alpinism can be defined as an endurance sport that requires certain moments of high intensity efforts, performed in a harsh environment and under moderate to severe hypoxic conditions. Mountaineering was born as an adventure and exploration activity restricted to a small number of people, but progressively became a common sporting activity. In the late 1970s, commercial expeditions began to operate at extreme altitudes mainly on Mount Everest, with progressively large numbers of amateur mountaineers attempting the summit of the Earth. This sport has evolved into a professional practice, and many athletes with severe training regimes perform yearly at high altitudes to summit 8000 m peaks self-sufficiently [2].

Extreme altitude strongly influences nutrition on expeditions, either because of the physiological challenges or the logistic aspects. Current sea level recommendations for nutrition in endurance sports are not realistic in extreme altitude expeditions, most notably when talking about alpine-style expeditions, so they need to be adjusted to real life conditions. This report aims to summarise the scientific knowledge around hydration, nutrition, and metabolism at high altitudes and to put them into the practical context of extreme altitude alpinism, which, as far as we know, has never been considered before in the literature.

### 1.1. Environmental Considerations

When considering nutrition recommendations for extreme altitude alpinism, it is mandatory to understand the context in which this sport takes place and the environmental limitations, which will be exposed along the following lines.

The playing field where alpinism takes place is located in a changing environment, which implies different conditions for the same objective that in turn modify the energetic requirements and performance [3]. It is worth considering that alpinism requires technical training in more than one specific sports modality and an adequate endurance capacity that allows to sustain their highly specific technical demands during a prolonged period of time (weeks, sometimes months). The tactical aspects in high-altitude alpinism involve choosing the route to the summit according to the characteristics of the athlete or the group of athletes, deciding the daily schedule according to the mountain conditions and the desired rhythm of ascent. This approach requires maintaining a delicate balance between high-altitude acclimatisation (in order to minimise the risk of high altitude illness and its severe life-threatening forms) and the time of exposure to severe altitude hypoxia (which involves de-training and a progressive deterioration of physical and mental performance, body composition, and health status).

The accurate calculation of energy expenditure in the field is a complex task. Three studies have measured energy expenditure at extreme altitudes using doubly labelled water, showing values between 3250 and 4636 Kcal per day (see Table 1). Other studies examining energy expenditure at lower altitudes show greater values (around 5000 Kcal) according to the theory of the economisation of oxygen consumption for metabolism as altitude increases, and the limitation of physical performance related to environmental hypoxia and the subsequent decrease on consumed energy (see Table 1). Other factors such as cold, the load carried [4], and the characteristics of the path [5] also influence the energy expenditure.

At sea level, the barometric pressure hovers around 760 mmHg, but at the summit of Mount Everest this value is reduced by two thirds, although the percentage of oxygen molecules in the air remains the same (20.9%). This atmospheric pressure drop has an impact on the oxygen availability, as the gradient of the partial pressure of oxygen diminishes from the inspired gas in every point of the cascade of oxygen transport from the lungs to the cells. Consequently, there is a lower flow of oxygen to the mitochondria, causing a shortage in aerobic ATP production, which impairs performance and can have an impact on health. Adding to the impairment of oxygen delivery, the energy requirements for a given task increase due to the hypoxic hyperventilation response, thermoregulation requirements, and the sympathetic activation secondary to hypoxia exposure. There is a specific altitude, which varies among individuals, where the detrimental effects of exposure are severe and acclimatisation is no longer effective. In this last point, athletes have the hardest times exercising in an extreme environment with very little aerobic capacity, no hunger, and great limitations for hydration [10].

An adequate basal aerobic capacity is needed in order to mitigate the effects of hypoxia regarding performance [11], as a reduction in the maximal oxygen consumption ($\dot{V}O_{2}\max$) of 1.5–3% for every 300 m of ascent is observed above 1500 m (although reductions have been observed from 580 m) [12], being a difference more marked with increasing altitude especially above 6300 m [13]. From this altitude level, the decline is even more rapid, probably linked to a reduced blood flow and the deterioration associated with chronic altitude exposure. An illustrating example of this is that the $\dot{V}O_{2}\max$ of an acclimated climber on the top of Everest is 20% of that measured at sea level, this decrease being variable between individuals but more marked in those highly fit [14].

Counterintuitively, submaximal exercise at extreme altitude does not imply greater oxygen consumption, probably because the need for acclimatisation does not allow one to reach the same fatiguing intensities that are reached at sea level. However, considering the decline in the $\dot{V}O_{2}\max$, the relative cost of performing at a same specific intensity increases with elevation. Anyway, it is difficult to predict the magnitude of the differences in climbing or walking performance at varying altitudes regarding the influence of other factors such as motivation, technical abilities, biomechanical efficiency, experience, capacity of tolerating mental stress and thermal discomfort in a certain moment [15].

Adding to the previous statement, a practical approach to nutrition, hydration, and survival at extreme altitudes has to consider that each mountain range and even each mountain in particular has its particularities regarding the possibility of food supplies, shelter, distance to the base camp, and rescue availability (see Table 2).

### 1.2. Style of Ascent

The style of the expedition has a strong influence on the energy requirements, closely related to logistics and technical equipment of the route (walking, climbing, and skiing have different energy requirements). Usually, those commercial expeditions with more economic support have hired staff who are responsible for cooking and obtaining potable water, arranging amenities (cooking room, shower, dining room, individual and collective tents), and providing the technical support to progress on the mountain (fixing ropes and stairs, protecting dangerous traverses, carrying supplementary oxygen supplies, and often playing a leading role in the decision-making as professional mountain guides). This support has a strong value as it provides time to rest after the strenuous physical efforts of climbing and supresses the psychological stress of making logistical and safety decisions. In addition, virtually all commercial mountain routes are carried out along well-known paths, generally previously traced, with predictable rates of ascent, foreseeable dangers, identified black points, places to camp, and scheduled times needed to reach the summit safely [16].

Conversely, professional alpinists tend to be self-sufficient in the mountains, which implies time and effort to cover not only the basic needs, nutrition, and hydration, but also to take care of their environmental survival and safety. This requires the technical capacity to solve the difficulties of the route for themselves.

It is also worth considering that the ascent profiles of these two different ways of climbing the same mountain have wide differences. While commercial ascents are slower, more time is spent at high altitudes (although supplementary oxygen is used at high altitudes), have more comfort for everyday needs, and require lower fitness, technical levels, and climbing experience, alpine ascents tend to cover longer distances for a given time, which increases energy expenditure whereas less time is spent at high altitudes.

Although logistics and ascent profiles can widely vary from one expedition to other, Figure 1 illustrates the main differences between both styles for a given route.

The most recognised alpinism awards consider as meritorious those non-previously explored routes, reaching the summit using less equipment or support while climbing [17], with a “clean sport” optic that refuses the use of preventive drugs (diuretics, corticoids, and sleep inductors are widely used in mountaineering [18,19]) and also, but not always, supplementary oxygen to achieve its sporting objective. The rationale of this point of view is far away from the objective of this text, but it is clear that adding the physiological stress of altitude to the intrinsic fatigue derived from accumulating all the alpine and carriage effort for itself, strongly modifies the energetic requirements, the nutritional needs, logistics strategy, and risk management. As both approaches are not physiologically comparable, from here, when talking about nutrition in this text, we will refer to high-altitude alpinists who perform in a pure alpine style, without the use of preventive drugs and supplementary oxygen.

### 1.3. Impact of High Altitude on Physical Performance

Early physiological responses to acute altitude exposure include a variety of compensatory adjustments that comprise an increase in the sympathetic activity with an increased heart rate, cardiac output, and blood pressure. The main sensors of hypoxia are located in the carotid bodies, which trigger the hypoxic ventilatory response, with higher respiratory rates and greater respiratory volumes leading to an increase in the minute ventilation. This acute response has the objective of buffering the lack of oxygen pressure by increasing the alveolar oxygen content and has been widely described as beneficial for acclimatisation [20]. These early responses translate to a progressive improvement in the time to exhaustion in tests performed at moderate altitudes [21]. Interestingly, it seems that highly trained endurance athletes can respond to exercise with lower ventilation rates, a physiological response that is beneficial at sea level [22] but that can blunt performance at high altitudes [23]. For extreme high altitude climbers attempting to perform above 8000 m without supplementary oxygen, it seems that a high ventilatory efficiency (described as the sustainable ventilation that permits maintaining an acceptable SatO_2_ but limiting minute ventilation) would be beneficial. Obviously, this implies an optimal ventilation/perfusion match, thus reducing the work of breathing, water, and heat losses and allowing for a sufficient ventilatory reserve [24] while preserving arterial oxygenation.

During the first two weeks at altitude, there is a small decrease in the maximal heart rate but a more pronounced reduction in the stroke volume and maximal cardiac output, which might give some kind of myocardial protection to hypoxia and limits an exaggerated energy expenditure [25].

These early responses are effective as an emergency mechanism but energetically expensive and are followed for more sustainable changes over time. The main effective changes in chronic acclimatisation to hypoxia are regulated at the molecular level by the hypoxia inducible factor (HIF), conformed by two subunits (α and β) which dimerize in the cell nucleus and activate hundreds of target genes involved in systemic and cellular adaptation to hypoxia. The genetic reprogramming induced by the HIF includes increased erythropoiesis, angiogenesis, upregulation of glycolytic enzymes, and inhibition of oxidative phosphorylation [26]. The increase in erythropoiesis leads to ameliorated values of submaximal endurance performance and $\dot{V}O_{2}\max$ [27]. See Table 3 for details about the pathophysiological effects of acute and chronic hypoxia.

It is noteworthy to consider that the reduced air density caused by low barometric pressure at the altitude could give some advantage (reduced aerodynamic resistance) in short and quick workouts with a main anaerobic component, rarely seen in alpinism.

## 2. Methods

Relevant literature regarding nutrition and high-altitude expeditions are scarce. We performed a search in the PubMed database. To optimise the identification of relevant articles, the terms “nutrition”, “hydration”, “metabolism”, “high altitude”, “mountaineering”, and “alpinism” were combined with Boolean operators (“AND” and “OR”) and searched from inception up until 4 June 2022. Filter options were applied in order to narrow the results regarding papers related only to human species, published in English, and with publication dates ranging between 1980 and 2022. A further selection was performed applying the following inclusion criteria: (1) randomised controlled trials, (2) pilot studies, and (3) clinical trials.

Two investigators (AC and JC) selected the eligible articles based on the title, abstract, and full paper, using the inclusion criteria. Disagreements were resolved by a consensus. The final selection of the studies was according to these phases: (1) identification of interesting studies, (2) duplicates removal, (3) title and abstract examination, (4) full text exploration, and (5) checking of the quality of the research and relevance with the purpose of the review.

## 3. Discussion

High altitude has a strong impact on muscle metabolism and substrate utilisation, so nutritional strategies should be designed in order to minimise the intrinsic negative energy balance observed in altitude alpinism.

Skeletal muscle is mainly an oxidative tissue dependent on oxygen to maintain energy homeostasis. Under normoxia conditions, oxidative phosphorylation can normally meet the muscular requirements of ATP. At altitude, a metabolic switch is needed to compensate for the reduction in the environmental partial pressure of oxygen. The HIF mediated response has many consequences on muscular metabolism: downregulation of mitochondrial biogenesis, upregulation of mitochondrial autophagy, and upregulation of pyruvate dehydrogenase kinase. The translation of all these phenomena is the attenuation of oxidative processes in contrast with a maintained total capacity for glycolysis [28].

The pathways involved in the transport of bicarbonate, hydrogen ions, and lactate are up-regulated in hypoxia, augmenting the dynamics of the acid–base balance, with an increased muscular buffer capacity [29,30].

Muscle lactate production at high intensities is blunted, a phenomenon named the “lactate paradox”, which includes a low accumulation of blood lactate in $\dot{V}O_{2}\max$ tests, especially in natives and lowlander acclimated subjects after chronic exposure. The precise mechanisms have not been fully elucidated but are influenced by changes in the lactate production from the muscles and lactate uptake by tissues [31]. It has been proposed that the sparing of glycolytic enzymes from general muscle wasting could justify this lower accumulation of blood lactate [32,33].

Muscular strength and maximal muscular power are conserved at altitude as long as the muscle mass is preserved [34,35], which is one of the challenges of high-altitude alpinists. There is a global reduction in the size of the muscle fibres (approximately 17% in exposures above 8000 m) but not of the total number, as a consequence of multifactorial etiology that involves detraining while acclimatising (for the same hypoxic stimuli, subjects who actively climb lose less muscle mass [36,37]), changes in protein metabolism and turnover, negative energy balance, gastrointestinal issues, and a lack of ingestion of high-quality protein.

Acute hypobaric hypoxia decreases leucine turnover and its uptake from the muscles [38], which has shown to cause the suppression of protein synthesis [39], and is associated with the downregulation of the mammalian target of rapamycin (mTOR) independently from food intake [40], suggesting that although nutrition may mitigate muscle mass loss, it would not completely prevent muscle wasting [41,42].

This cachectic response may bring some kinds of benefits regarding an increased ratio of capillaries to muscle fibres cross sectional area [43]. In this regard, muscle wasting at altitude would benefit the delivery of oxygen to the mitochondria by decreasing the diffusion distance, but sacrificing maximal aerobic performance and strength, so increasing the risk of prolonged exposure and injury [44].

Regarding substrate utilisation, measurements of the respiratory quotient show the downregulation of fatty acid metabolism [45] with a preferential use of carbohydrates, which is more efficient in terms of energy yield per unit of O_2_ consumed either in trained or sedentary individuals [46,47]. Hypoxia stimulates the glycolytic flux and increases the availability of pyruvate, due in part to the elevated levels of epinephrine.

Moreover, carbohydrate use as the preferential energy source has additional advantages: a high carbohydrate, low fat diet at altitude moves the respiratory quotient (RQ) from 0.7, if exclusively fat is used for energy production, to near 1 when carbohydrate (or protein) is used. A consequence of such a variation in the RQ is that at any given Paco_2_ the Pao_2_ is increased, thus providing a substantial gain in arterial oxygen saturation. In a field study, forty-one members of a 22-day high-altitude expedition were randomised in a double-blind design to receive either placebo or carbohydrate supplementation. Participants performed a mountaineering time trial at 5192 m. Subjects of the carbohydrate-supplemented group reported 18% lower ratings of perceived exertion during the time trial at altitude and completed it 17% faster than the placebo group. However, cardiovascular parameters obtained during submaximal exercise and spontaneous physical activity on rest days were similar between the two groups [48]. A second reason for recommending a high carbohydrate diet is that it has been suggested that the body becomes more dependent upon glucose as a fuel at altitude as acclimatisation progresses [49]. When the same absolute rates of exercise were compared, higher glucose use and fewer free fatty acids were detected, but this change faded when comparing the same relative rates of work. It could be inferred that, in practice, climbers at altitude must often work at higher relative rates than at sea level, even though that means a lower absolute work rate. Therefore, it seems logical that it should be carried out with a higher use of carbohydrates.

Lipolysis during post-exercise recovery also seems to be impaired. Although the concrete mechanisms are still unknown, it is suggested that higher insulin concentrations at altitude would inhibit lipolysis, and that the stimulation of glycolysis would impair free fatty acid mobilisation and that suppression of the peroxisome proliferator-activated receptors (PPAR) would blunt lipid oxidation [50].

### 3.1. Body Weight Loss and Changes in Body Composition at Altitude

The loss of appetite and weight are very usual at altitudes and can be one of the symptoms of high-altitude sickness. Anorexia at altitude is nearly universal, hormonally mediated by the actions of leptin, ghrelin, and cholecystokinin [51,52].

Body weight loss depends on the duration of the exposure and the altitude reached [53,54] and results in changes in the body composition (see Table 4). Alpinists lose primarily fat and then protein at higher altitudes in a context of chronic dehydration. Above 5000 m, 60–70% of weight loss comes from fat-free mass [7,55]. At lower altitude, the weight loss has been mitigated by providing food matched to the energy expenditure [56] or with highly palatable food ingestion [57]. These results have not been replicable at higher altitudes, where reduced food intake and weight loss were observed despite food availability and palatability [58]. 

The majority of studies revealed that high-altitude-induced weight loss is the consequence of a negative energy balance produced by an inadequate energy intake, below 50–70% of the daily requirements [59]. This does not match the increase in the basal metabolic rate (10–28% at 4000–6000 m) secondary to the work of breathing and thermoregulation required in the extreme environment and the increase in energy expenditure compared to sea level [60]. In this regard, although the energetic requirements are high, one must assume that at high altitudes the energetic balance will be negative due to a series of unsurmountable difficulties to cover the energy intake caused by altitude anorexia, altered metabolism, the fatigue associated with carrying large amounts of food and supplies for cooking (which cannot be faced at extreme altitudes by non-commercial, sporting expeditions with little or null porters support), and difficulties in melting the required quantity of water for drinking, rehydrating food, or cooking. Finally, we must consider the poorly understood mechanisms that underlie malabsorption and gastrointestinal distress [61].

**Table 4 ijerph-20-03186-t004:** The changes in body composition at altitude, adapted from Wing-Gaia, 2014.

Reference	n	Altitude (m)	Time at Altitude (days)	Total BW Loss (kg)	Body Fat Loss (kg)	FFM Loss (kg)	% BW Loss (FFM)
[55]	10♂	≥4500	16	3.3	2.2	1.1	33.3
	2♀						
[62]	14♂	<5400	23	1.9	**1.34**	**0.56**	**29.5**
		>5400	26	4	2	2.8	70
[45]	16♂	3700–4300	16	5.9	2.53	3.37	57.1
[6]	5♂	5900–8046	40	3.7	0.9	1.9	51.4
	1♀						
[36]	8♂	8846	38	7.4	2.51	5.05	66.8
[63]	5♂	2200–3400	21	4.2	1.1	3.2	75
[8]	3♂	5300–8872	30	2.2	1.4	0.8	36.4
	2♀						
[60]	4♂	6542	21	4.9	3.5	1.3	27
	2♀						
[64]	10♂	2835–5364	13	* C:1.8	0.6	1.2	66
	8♀			L: 1.9	1.1	0.8	42

BW: Body weight. FFM: Fat-free mass. Men (♂) Women (♀). * C: control group; L: Leucine supplementation group.

### 3.2. Macronutrient Needs at High Altitude

There is no current bibliography available about macro and micronutrient needs at extreme altitudes, mainly because of the difficulties in conducting valid experimental designs and the limited number of athletes who perform at such severe levels of hypoxia.

There is strong evidence regarding the preferential use of carbohydrates as fuel by the working muscles, which is in accordance with the observation that in intense training sessions at altitude, carbohydrate intake among alpinists can increase up to 80% of the total calories per day [9].

As previously described, carbohydrate ingestion before exercise in simulated hypoxia helps to maintain oxygen saturation and ventilation [65] and glucose administration during prolonged activity at high altitude is beneficial for performance [66], although this effect seems to be more intense in the first days of exposure and diminishes as acclimatisation takes place [67].

It must be considered that acute exposure to hypoxia (<8 h) impairs the oxidation of exogenous carbohydrates, partially due to an hyperinsulinemic state, with a corresponding increase in the endogenous carbohydrate utilisation (blood glucose, muscle, and liver glycogen), that reverts with acclimatisation in spite of the weight loss [68]. In this regard, efforts should be made to face the expedition with an optimal development of glycogen stores, to be able to perform in the early phase of the acclimatisation, when a supplementation of carbohydrate intake would be of little benefit.

Regarding current guidelines, in order to maintain health and sustain performance, athletes should provide the right amount of carbohydrates before, during, and after exercising in hypoxia, as to preserve glycogen stores, enhance muscular recovery [69], and ensure immune activation [70].

At sea level, experts recommend 8–12 g of carbohydrates per kilogram of body mass per day for reaching an athlete’s basal daily fuel needs [71] and additionally 30–70 g of carbohydrates per hour of exercise depending on the exercise intensity and duration, or even higher for ultra-endurance events [72,73]. At high altitudes, the gastrointestinal issues, lack of appetite, and logistics make these recommendations unreal. So, athletes should try to approximate as close as possible to them by selecting appropriate (nutrient dense, chewable, palatable) carbohydrate-rich foods when planning the expedition and try to eat as much as possible while exercising as long as they do not have limiting symptoms (see Table 5). Considered this limited capacity for food intake, it seems intuitive to respect the metabolic window after exercise as an opportunity to refill the glycogen stores, when the long-term carbohydrate replacement is uncertain or there are limited food supplies, but research is needed in this aspect. Whether gut training [74] while yearly training and pre-acclimatising can be a potential key to improve food ingestion tolerance at altitude has not yet been studied.

From a practical point of view, it is worth considering that if the ambient temperature is low, some carbohydrate sources such as sports bars are not possible to chew and eat, and that liquid forms get frozen if not maintained near the body surface. Presumably, due to their high content of sugars and other additives, sports gels do not freeze easily, but may offer variable gastrointestinal tolerance by athletes at high altitudes.

Considering the need for fat-free mass retention at altitude, one may consider that increasing the ingestion of protein while exercising at altitude could help. Nevertheless, this hypothesis is limited by various constraints: (1) the problem of the conservation of animal and high biological value sources of protein, (2) the satiating effect of protein which could reduce carbohydrate intake [75], (3) the thermogenic effect of digesting protein, which in hypoxic conditions could mean an excess of energy cost [76], (4) although there are contradictory reports, some degree of malabsorption has been described which could illustrate the clinical issues found in some climbers who are not able to eat protein at high altitudes [77].

Considering that a diet with more than 2.0 g of protein per kilogram of body weight per day did not protect lean body mass loss during energy deficits at a high altitude [78], and that protein intakes in the range of 1.3–1.8 g per Kg per day distributed along the day in frequent meals maximise the protein synthesis at sea level [79], recommendations for protein ingestion at altitude should be made considering the ingestion of high quality protein as such to reach a minimum 1.3 g per Kg per day as long as it does not impair carbohydrate ingestion, with a careful timing distribution and prioritising key branched-chain amino acids such as leucine, which can act either as a substrate and as a regulator of protein synthesis [80].

Despite their greater caloric richness, foods rich in fat are not appetizing for most climbers. In addition, the tolerance to fatty foods is low at altitude and is often associated with gastrointestinal symptoms. Athletes should be advised to eat fat in the form of unsaturated oils such as olive or sunflower for cooking and dressing and from those contained naturally in nuts and animal sources of protein (see Table 5).

### 3.3. Micronutrient Needs at High Altitude

As a result of erythropoiesis in acclimatisation, depletion of iron storage is observed. Low levels of iron and ferritin can impair the increase in haemoglobin concentration. Due to the slow capacity of replenishment, iron storages must be fully replenished before the expedition [81]. There is no evidence of the benefits of iron supplementation at altitude if pre-expedition ferritin levels are correct. Blood hyperviscosity due to excessive haematocrit increases the risk of high-altitude thromboembolism, a fact that must be considered when bearing in mind the adequacy of iron supplementation.

No information is available regarding micronutrient needs at extreme altitudes [82], but considering the lack of fresh food and protein sources, an interesting aspect would be to consider whether alpinists have deficits in vitamins B, C, E, or other elements such as polyphenols or nitrates that should be replaced during long stages and could justify supplementation. Antioxidant elements may contribute to mitigating the impact of oxidative stress related to the altitude. Although hypoxia-induced free radical formation may contribute to molecular signalling to produce the acclimatisation to altitude, if produced in excess, muscle function and capillary perfusion can be impaired. The threshold from which oxidative stress becomes a burden is not yet well understood, but it is interesting to consider dietary or supplementary antioxidant sources as a tool when altitude exposure leads to an important stress in the body, particularly if combined with fatigue and thermal stress [83].

### 3.4. Hydration at High Altitude

Adequate hydration is crucial regarding performance and health. Athletes at extreme altitudes are at risk of dehydration because of the high respiratory water loss due to the increased ventilation and an increased urinary loss secondary to the downregulation of the renin-angiotensin-aldosterone system. Other than this, the effect of a dry air environment that favours perspiration cannot be neglected. Regarding the data, at moderate altitudes (up to 4000 m) respiratory water loss may be increased up to 1900 mL in men and 850 mL in women per day, together with an increase in 500 mL of urinary water loss per day [84].

Alpinists use glacier water as the primary source for hydration and cooking, which is low in minerals and electrolytes. It is worth considering the need for salt and electrolyte supplementation to ensure correct hydration; although the exact quantities must be individualised, as general recommendations during mountain activities consider drinking 400–800 mL/h with 0.5–1 g Na/L of water. This statement should be adjusted to individual sweat rates, urinary control, and symptoms of altitude illness, as this pathology causes fluid retention. Hot drinks are encouraged in cold environments to prevent hypothermia and as a palatable source of hydration in the form of tea or herbal infusions.

Alpinists should be aware that hydration is one of the main preventive measures against frostbite, ideally with diluted carbohydrates to maintain the shivering capacity in order to prevent moderate to severe hypothermia.

The main handicap for hydration is melting ice or snow at altitude. The water boiling point decreases with altitude, but extremely low ambient temperatures and quick evaporation make the procurement of water extremely difficult, considering that the salt content in the natural sources of water is usually very low at altitude. It is worth considering that food needs more time to be cooked as boiling temperatures are lower. Although this process has been ameliorated with the use of vacuum water boilers, this task requires all the resting time and energy of alpinists at high altitude and it is one of the main constraints for survival and keeping of the athletic performance.

### 3.5. Nutrition in the Different Stages of an Expedition: Trekking and Acclimatisation to Moderate Altitude

This phase involves travelling from the lowland to the mountain range chosen for acclimatisation. This is normally completed by an approach hike to base camp at altitudes up to 4500 m.

It is not uncommon that an important part of the food that will be consumed during the expedition is bought at the destination. At this point, the cultural differences regarding food and cooking have to be considered; in the case of hiring the services of a local cooker, it is important to supervise the procurement of the food and the spices. Added to this fact, religious aspects can determine the availability of animal protein sources. In Buddhist countries such as Nepal, obtaining meat can be more challenging than in other places, especially in small valleys or remote areas (see Table 2 for the particularities of each mountain range and Table 6 for the adequate carbohydrate and protein sources for each different stage of the expedition). Infectious diarrhoea is a very common issue at this point, which can be prevented with general recommendations of food conservation and hygiene but also with being cautious when eating in local restaurants, eating street food, or drinking water.

In this phase, athletes normally choose to stay in high-altitude villages sited in valleys offering relatively comfortable commodities such as a heated room, food, and sleeping in a bed. Normally there is no problem with appetite and digestion related to altitude, so efforts should be made in order to sustain adequate intakes adjusted to the workouts.

The time needed for training comfortably at this altitude varies among individuals and can be conditioned by the habit of the athlete to train at altitude or if there has been a pre-acclimatisation programme at home.

Some authors have suggested that the ingestion of antioxidant-rich foods such as fruits, berries, nuts, dark chocolate, etc. increase the antioxidant capacity in athletes performing at moderate altitudes, but with no impact on oxidative stress [85]. Considering that exogenous antioxidants neutralise free radicals with no known adverse effects on health, it seems logical to consider that antioxidant supplementation would be an interesting tool against altitude-induced oxidative stress during acclimatisation, where no muscular adaptation is searched for that can be blunted with this intervention [86]. As findings in this regard are not yet consistent, athletes at altitude should be encouraged to consume antioxidant-rich foods as part of the diet, but supplementation does not have sufficient evidence yet.

### 3.6. Nutrition in the Different Stages of an Expedition: Base Camp

Base camps are usually located between 4500 and 5500 m of altitude.

For athletic expeditions, where logistics are different and normally more restrictive than in touristic or commercial expeditions, once arriving to base camp supplies cannot be replenished until the expedition ends. Alpinists usually carry all supplies with them to stay for weeks, so there is the need for a weight and volume strategy to transport the maximum amount of quality and palatable food. Trying to avoid monotony of the diet is hard work but helps to motivate athletes to eat.

Hygiene has to be highlighted as an important issue for those attending the kitchen. Other strategies include locating latrines far from water sources and living zones, drinking and conserving properly potable water, and avoiding the presence of animals around food supplies.

One important limitation is the storage and conservation of fresh vegetables and fresh fruits, as well as animal protein sources, which mainly come from chickens and yaks but tend to be less abundant as the expedition goes on. Fish sources are only eaten as canned food.

This is the last point of the expedition where athletes can have a complete diet, based in fresh and cooked foods. It is important to consider that although acclimatisation continues and physical activity is supposed to be sustained, and is encouraged, in order to preserve a lean body mass (although at lower volumes and intensities than at home), there are long journeys where alpinists are not able to go outside their tents because of the weather, so energy expenditure can vary a lot from one day to the following.

On the other hand, supplementation with branched amino acids can be considered if there is no possibility for matching protein needs, although supplementation has not demonstrated benefits in preserving lean mass [87,88].

From this point, water supplies come from melting water of the glacier. Supplementation with salts and electrolytes is mandatory for all drinking water, as they are not naturally found in snow or ice at altitude.

### 3.7. Nutrition in the Different Stages of an Expedition: High-Altitude Camps

Reaching high-altitude acclimatisation is a difficult task and is conditioned by individual aspects, weather, and mountain conditions, as seen above. Strategies for facing summits above 8000 m vary a lot and comprise different acclimatisation profiles (often mixed): from gradual climbing with the installation of intermediate camps, to relatively quick ascents, to progressively more altitude but returning to base camp to recover and sleep (see Figure 1). Normally, dehydrated or lyophilised foods are consumed widely in this phase, and the loss of appetite and difficulties in melting water, cooking food, and carrying fuel and supplies are challenging and worsen with the altitude (specially above 7000 m). Preferences for foods at high altitude vary a lot among climbers and usually change as the altitude increases.

Athletes must be informed that their appetite might not reflect the real nutritional needs; they should be encouraged to eat and drink as much as they can but maintain the quality of the food. In this regard, dense, small, and frequent meals and easy to eat snacks can be useful to reach the higher energy requirements under these conditions.

### 3.8. Nutrition in the Different Stages of an Expedition: Summit Day

At this point athletes accumulate fatigue and deterioration related to altitude and have to face the most challenging aspect of the expedition regarding altitude and cold with a strongly diminished aerobic capacity. Reaching the summit and returning to the high-altitude camp can take from 12 to 24 h, although this estimation can vary a lot. It is clear that there is little chance of survival if the exposure to such extreme altitudes is prolonged by any circumstance.

The capability for transporting water and food supplies is extremely limited and athletes perform with heavy equipment in order to face extreme cold temperatures, so there are few opportunities to eat due to the complexity of manipulating pockets and food with gloves or to stop the ascent in order to eat. Water tends to freeze if not located inside the clothing protection, close to the body surface. Dehydrated or solid food preparations, such as sports bars, are impossible to eat and swallow, as salivation is extremely lowered and chewing represents an extra effort. The most widely used strategy at this stage is to eat small portions of easy-to-digest shots of energy, such as gels, chocolate, or carbohydrate jellies with one or one and a half litres of available water, usually with a diluted carbohydrate solution.

## 4. Conclusions

The magnitude of the impairment of performance under hypoxic conditions is closely linked to the aerobic contribution of the work required. Lowering oxidative metabolism, and improving the bioenergetics efficiency and ventilatory response are the key mechanisms for optimal acclimatisation in regard to performance. There is little information about nutrition at extreme altitudes due to the environmental limitations for field studies and the small number of subjects who practise this modality of alpinism. Current sea level recommendations are not realistic in extreme altitude expeditions, most notably when talking about alpine-style expeditions, so they need to be adjusted to real life conditions. Requirements for the different stages of an expedition vary dramatically with increasing altitude, so recommendations must differentiate whether the alpinist is at base camp, at high-altitude camps, or attempting the summit. General recommendations include prioritising carbohydrates as a source of energy, while trying to maintain a protein balance during the expedition and a correct hydration status. More research is needed regarding the specific macro and micronutrient requirements as well as the adequacy of nutritional supplementation at high altitudes.

## Figures and Tables

**Figure 1 ijerph-20-03186-f001:**
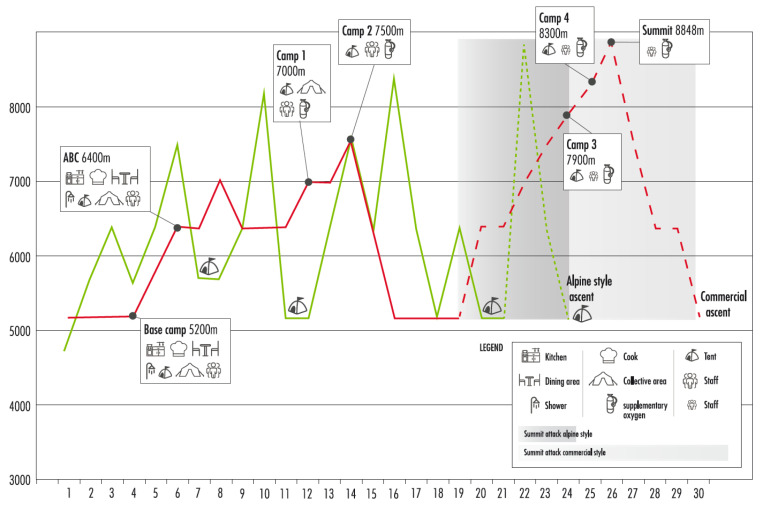
The typical acclimatisation and ascent profiles of commercial and alpine-style ascents to Mt. Everest (North face).

**Table 1 ijerph-20-03186-t001:** The energy expenditure in alpinism.

Mountain Range	Altitude (m)	Duration (days)	Energy Expenditure (Kcal/day)
Alps [4]	3422	2	5572.8
Himalaya [6]	5900–8046	7	4636
Himalaya [7]	8840	7–10	3250
Himalaya [8]	8840	40	3274 climbers/5394 porters
Cascade Range [9]	2500–3100	7	4558

**Table 2 ijerph-20-03186-t002:** The particularities and practical aspects of each 8000 m peak.

Mountain	Altitude ^1^	Country	Mountain Range ^2^	Capital of Arrival ^4^
Everest (Sagarmāthā/Chomolungma)	8848 m	Nepal/China	Himalaya	Kathmandu
K2	8611 m	Pakistan/China	Karakorum	Islamabad
Kanchenjunga	8586 m	Nepal/India	Himalaya	Kathmandu
Lhotse	8516 m	Nepal/China	Himalaya	Kathmandu
Makalu	8463 m	Nepal/China	Himalaya	Kathmandu
Cho Oyu	8201 m	Nepal/China	Himalaya	Kathmandu
Dhaulagiri	8167 m	Nepal	Himalaya	Kathmandu
Manaslu (Kutang)	8163 m	Nepal	Himalaya	Kathmandu
Nanga Parbat (Diamer)	8125 m	Pakistan	Himalaya ^3^	Islamabad
Annapurna	8091 m	Nepal	Himalaya	Kathmandu
Gasherbrum I/Hidden Peak	8068 m	Pakistan/China	Karakorum	Islamabad
Broad Peak	8047 m	Pakistan/China	Karakorum	Islamabad
Gasherbrum II/K4	8035 m	Pakistan/China	Karakorum	Islamabad
Shishapangma (Gosainthān)	8027 m	China	Himalaya	Kathmandu

^1^ Altitude strongly influences the technical difficulties. The highest mountains (Everest and K2) have two main problems: the base camp and last night bivouac before the summit are located at a higher altitude. ^2^ The climate is different among the two mountain ranges. The Himalaya usually follows the monsoon cycles whereas the Karakorum has a continental climate (North hemisphere). ^3^ Although it belongs to the Himalaya, it is located in the occidental border of the range and has a continental climate. ^4^ In Kathmandu all food supplies are available, including lyophilised food. Nepal’s population is mostly Buddhist, so they eat little meat. In the trekking paths there are lodges. From the base camp, porters can go villages to buy greens. In Islamabad there are food supplies although finding sports food can be difficult. The population is mostly Muslim, so they eat meat and live animals can be transferred to some base camps. Greens are far from base camps and the paths of the trekking do not have lodges, so climbers must be self-sufficient from the beginning of the approximation to the mountain.

**Table 3 ijerph-20-03186-t003:** The pathophysiological effects of acute and chronic hypoxia.

System	Acute Exposure	Chronic Exposure	Effects on Performance	Strategies
Pulmonary	↑ ventilation ↓ arterial oxygen saturation	Hypoventilation ↑ lung capillary blood volume pulmonary hypertension	Increased cost of breathing. Irritative cough at altitude due to dehydration of mucosa is nearly universal.	Gentle acclimatisation. Protecting face with masks that maintain humidity in the bronchial tract.
Cardiovascular	Transient ↑ blood pressure ↑ heart rate ↑ cardiac output,	↓ systolic & diastolic pressure,	Decrease in physical performance. Difficulty to measure the intensity of exercise by using the heart rate.	Optimal strategy of acclimatisation.
Hematological	↑ haemoglobin concentration ↓ plasma volume	Polycythaemia ↑ blood O_2_ carrying capacity	Helps performance at altitude, but needs time. Can be a cause of thrombosis among climbers.	Leave enough time for altitude exposure so erythropoiesis can begin. Hydrate and avoid long periods of inactivity when there is bad weather to prevent thrombosis
Renal	↑ bicarbonate excretion hypocapnic respiratory alkalosis ↑diuresis	Hyperuricemia microalbuminuria ↓ renal plasma flow ↑ filtration fraction	Partial compensation of the effects of hypoxia. Some drugs as acetazolamide can enhance this physiological response. Dehydration can occur.	Optimal hydration strategies and urine colour control.
Metabolic	↓ use of exogenous carbohydrate ↑ use endogenous carbohydrate ↓ use of fat	↑ glycolytic enzymes ↓ oxidative phosphorylation ↓ blood lactate ↑ buffer capacity ↑ carbohydrate as fuel	Not known, but theoretically more efficient in terms of oxygen cost for metabolism.	Provide enough carbohydrates in the diet.
Neurological	↓ synthesis of neurotransmitters, mood change, ↓ motor/sensory functions	Cerebral hypoxia, biochemical dysfunction ↓ sleep quality ↑ mood disorders	Strong impairment of motivation, lucidity, and quality of sleep.	Avoid sedative medication to prevent cerebral hypoxia while sleeping. Try to sleep at lower altitude to recover.

**Table 5 ijerph-20-03186-t005:** The nutritional recommendations for the different expedition stages.

Expedition Phase	Principles	Recommendations	Logistics
Trekking and moderate altitude acclimatisation	- Low training intensities, especially at the beginning of the acclimatisation. - At the beginning no training adaptation is searched. - During the first hours of altitude exposition, body might be unable to use exogenous carbohydrates. - No problems with food supplies and food tolerance.	- Athletes should face an expedition with optimal training and nutrition status. - Carbohydrate needs: 3–5 g/less/d. - Protein needs: 1.3–1.5 g/Kg/d - Consider eating food rich in antioxidants. - High attention in avoiding infectious diarrhoea. - Drink bottled water. - Eat complete meals including fresh fruits and vegetables, as well as high biological quality protein sources.	- Organise and buy the food needed for the whole expedition in the closest city. If travelling with an expedition agency, logistics are easy. - Planning the approximation to the mountain must include stopping at villages with bottled water availability - No logistic problems to include three main meals. - Special focus on proper water disinfection. - Avoid raw food, juices made with ice of non-controlled sources, peel the fruit yourself, animal food must be completely cooked. - All food should be cooked just before ingestion. - Clean raw vegetables with disinfectant solution.
Trekking and moderate altitude training	- Training intensity increases, but will not reach maximal values. - Need to restore glycogen storages in high intensity workouts. - Need to maintain lean body mass and prevent weight loss. - No problems with food supplies and food tolerance.	- Carbohydrate needs: 5–8 g/Kg/d. - Protein needs: 1.5–2 g/Kg/d. - (carbohydrate periodization depending on exercise intensity and acclimatisation status). - After workout: eat carbohydrate-rich foods with moderate–high glycaemic index to enhance glycogen storage, consider adding protein. - High attention in avoiding infectious diarrhoea. - Drink bottled water. - Eat complete meals including fresh fruits and vegetables, as well as high biological quality protein sources. - Consider casein ingestion before bed.	- Two main meals during the days of training and acclimatisation. - Snacking during training and ascents. - Design acclimatisation and training regarding lodge availability with food and bottled water supplies. - Ensure protein sources and correct timing of ingestion. - Maintain focus on hygiene.
Base camp	- Variable intensity workouts, but will not reach individual maximal intensities with the increasing altitude. - Carbohydrates are the main source of energy. - Limited food supplies. - Need to maintain lean body mass and prevent weight loss. - Risk of altitude sickness. - Hydration supports training and helps prevent cold injuries.	- Carbohydrate needs: 3–8 g/Kg/d - Protein needs: 1.3–2 g/Kg/d - Carbohydrate periodization depending on exercise intensity and acclimatisation status. - Try to have at least two complete main meals including fresh sources of protein, fruits and vegetables, and carbohydrates of low glycaemic index. - After workout: eat carbohydrate-rich foods with moderate–high glycaemic index to enhance glycogen storage, especially if energy intake is limited; consider adding protein. - Add salts and electrolytes to drinking water. - Monitor hydration status with urine colouration. Try to drink frequently, and as much as possible. - Consider leucine supplementation if problems achieving protein intake. - Provide fat mainly from olive or sunflower oil, and contained in meat, fish, or cheese.	- Previous scheduling of food supplies in order to diminish monotony of foods in base camp. - Ingestion must be adjusted to the work required regarding acclimatisation, meteorology and eventual preparation of the route and high-altitude camps. - In days of bad weather or inactivity, 3 main meals are recommended. If many days of bad weather, try to maintain nutritional status without gaining too much weight. - In days of physical activity, 2 main meals are recommended adding snacking during exercise. - Maintain focus on hygiene - Latrines must be far from the living zones. - No animals near water sources and food.
High-altitude camps	- Altitude strongly impairs exercise intensity. - High risk of illness related to altitude and cold. - Carbohydrates are the main source of energy. - Limited food supplies. - No fresh foods available. - Appetite decreases with altitude. - Unavoidable weight loss. - Hydration helps avoid cold injuries. - Risk of infectious diseases diminishes with altitude. - Apathy for cooking, eating, and hydrating increases with altitude.	- Try to have a complete breakfast and a complete dinner. - If no appetite, try to eat frequently small amounts of high energy dense food. - Add salts and electrolytes to drinking water. - Avoid food containing fibre that blunts absorption. - In high intensity days, consider using a recovery drink early after workout. - Monitor hydration status with urine colouration. Try to drink frequently (water, soups, infusions), and as much as possible. - Consider leucine supplementation if problems achieving protein intake. - Consider supplementation with vitamins if long stay at high altitude without fresh foods. - Provide fat mainly from olive or sunflower oil, and contained in meat, fish or cheese. - Maintain motivation for preparing food and melting water, or distribute mandatory tasks among members of the expedition.	- Special attention to timing of ingestion when limited sources of carbohydrates or protein are available. - According to expedition capabilities, communication with base camp for scheduling recovery meals when descending from acclimatisation. - Provide enough fuel for cooking. - Water is melted quicker in vacuum cooking pots designed to be light. - Consider packaging food for each day. - Start the day melting ice for breakfast, drinking water before starting to walk and to fill the bottles used during ascent. - Plastic or aluminium bottles freeze quickly, consider using hydration packs with short tubes protected from cold. - Bottles with thermal protection are useful but add weight. - First member arriving to the tent every journey starts melting water for rehydration of food. - If there is an excess of melted water, use it for infusions.
Summit day	- Very low exercise intensity. - Extreme risk of altitude and cold-related illnesses. - Carbohydrates are the only source of energy. - Fatigue and deterioration are the main challenges. - Appetite suppression. - Little capacity to carry water and food. - No risk of infectious disease.	- Consume hydrogels or gels that do not need considerable amounts of water to be ingested. - Force yourself to eat and drink, but consider the timing of the journey. - Limited ingestion might not need multiple carbohydrate sources. - Try tolerance to gels before summit. - Consider adding a carbohydrate drink mix to the water.	- Boil the water needed the previous day and conserve it inside the tent, as warm as possible. - During the journey, keep the food and the water in the inner pockets of the climbing suit. - If using hydration packs, try to make sure that the tube is not long and is correctly protected in order to not freeze. - Choose food that does not need the removal of gloves to be ingested. - Leave food and water for the descent or unexpected delays. - Try to have a previous idea on nutrition and hydration timing. Although changes might be needed, having a structured plan might reduce the impact of cognitive impairment at altitude in decision making.

**Table 6 ijerph-20-03186-t006:** The main carbohydrate and protein sources in high-altitude expeditions.

Expedition Phase	Carbohydrate Sources	Protein Sources	Practical Aspects
Base camp	Pasta, rice, quinoa, polenta, potatoes, sliced bread, *chapatis*, crepes, porridge, muesli/granola, chocolate (solid or soluble), jam, honey, dried fruits (dates, figs, apricots).	Eggs, meat (chicken, yak, goat), legumes, cheese, sausage, milk powder, ham, canned fish, nuts, textured soya, silken tofu.	- Expeditions may have cookers in the Base Camp. - No restrictions in the water supply. - No restrictions with boiling/warming times. - No restrictions in availability of time for cooking. - Gas and fuel restrictions linked to economic capability of the expedition. - Fresh fruits and vegetables only available during the first days/weeks of the expedition, depending on the location of the base camp. - Fresh meat only available during first days/weeks of the expedition, depending on location. - No fresh fish sources available, only canned.
Mountain activities and training during expedition.	*Chapati* or sliced bread, sports bars, dried fruits, mashed salty potatoes, sandwiches or wraps, rice or polenta cakes, chocolate. Carbohydrate and electrolyte-enriched beverage.	Eggs, ham, cheese in small amounts, nuts.	- No cooking opportunities during training. - Alpinists carry the food/water during training - Rationale between volume/weight and caloric content, high dense foods preferred. - Must be used to eating this kind of food during training at home. - No cutlery. - Variable eating tolerance while training at moderate altitude.
High-altitude camps	Easy to cook sources: couscous, polenta, instant potato pure, instant soup. Figs, dates, chocolate (better if soluble), jam, honey, cereal baby food, sliced bread, cookies. Muesli/granola. Lyophilised or dehydrated food.	Ham, sausages, cheese, previously boiled eggs, canned tuna or sardines. Milk powder. Lyophilised or dehydrated food.	- No cooker available. - Only instant foods that do not require preparation (ensure no problems caused by palatability/spices). - Water source from glacier. - Melting water takes time and effort proportional to the altitude. - Gas availability for melting ice or heating water is limited. No cooking is mandatory.
Summit	Sports gels. Gummy bars. Chocolate. Carbohydrate–electrolyte beverage. Rehydration solutions.		- Extreme limitation for water and food availability. - All food and water must be prepared the previous day and must include energy and hydration for climb and descent. - Small opportunities for ingestion. - Clothing equipment against cold makes it difficult to manipulate food and water. - Food or water carried out of the body freezes in little time. - Solutions rich in sugar suchvas sports gels do not freeze. - Sports bars, dry food, turn hard because of the extreme environmental cold and are not possible to chew and swallow.

## Data Availability

Not applicable.

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
