# Peer review of "Nutrition and Hydration for High-Altitude Alpinism: A Narrative Review"

_ijerph, 2023, doi:10.3390/ijerph20043186_

Round 1
Reviewer 1 Report (New Reviewer)
General comments
The aim of the narrative review was to summarize the scientific knowledge around hydration, nutrition, and metabolism at high altitude and to put them into the practical context of extreme altitude alpinism (>8000 m), which, as far as we know, has never been considered before in the literature. The manuscript is well-written and the topic of high interest for many readers but I think it would be valuable to modify somewhat the structure of the manuscript to highlight the novelty (nutrition) compared to what is already better described in the literature (physiology).
Major comments:
- Title: “Nutrition and hydration for high altitude alpinism: A practical 1 approach”. I’m not sure this is a practical approach as no concrete recommandations are made. I would suggest to modify “A practical approach” into “A narrative review”.
- Clearly this is not a systematic review nor a meta-analysis, it is thus unclear why the authors used the PRISMA guidelines. This is a non-sense. Certainly as those guidelines are not exploited further in the manuscript. For example, the quality of the studies is not reported, which is not a problem to me but this is recommended for a systematic review. There is thus inconstancy between the reported methodology and the structure and content of the manuscript.
- The objective of the work is to summarize the scientific knowledge around hydration, nutrition, and metabolism at high altitude and to put them into the practical context of extreme altitude alpinism (>8000 m). I don’t understand why the authors specify the altitude here as the articles have not been selected on that parameter. Many articles deal with lower altitudes. I would suggest not to specify the altitude in the objective of the work.
- lines 11-125: “Anyway, it is difficult to predict the magnitude of the differences in climbing or 122 walking performance at varying altitude, as there are other factors as motivation, technical 123 abilities, biomechanical efficiency, experience or the capacity of tolerating mental stress 124 and thermal discomfort that cannot be measured (7,11).” This statement is not correct as each variable mentioned here can be quantified by physiological tests and/or questionnaires.
- Maybe my most fundamental comment is the confusion between the original/novel input (nutrition/hydration) and the already known (physiology). Everything dealing with physiological adaptations is already well described in the literature compared to nutrition and hydration. To me, the title reflects the novelty as it only mentions nutrition and hydration. Then, the objective is probably a bit less novel as it proposes to summarize the knowledge around metabolism, which is already treated in the literature. I would suggest to move everything that deals with physiology and performance (lines 68 to 262) to the introduction as it is not specific to the new input to the literature and does not respond to the key words used for the search strategy.
Author Response
General comments
The aim of the narrative review was to summarize the scientific knowledge around hydration, nutrition, and metabolism at high altitude and to put them into the practical context of extreme altitude alpinism (>8000 m), which, as far as we know, has never been considered before in the literature. The manuscript is well-written and the topic of high interest for many readers but I think it would be valuable to modify somewhat the structure of the manuscript to highlight the novelty (nutrition) compared to what is already better described in the literature (physiology).
Thank you for your kind comments. We have rearranged a great part of the manuscript (that was not marked in red when text was not altered)
Major comments:
- Title: “Nutrition and hydration for high altitude alpinism: A practical approach”. I’m not sure this is a practical approach as no concrete recommandations are made. I would suggest to modify “A practical approach” into “A narrative review”.
Thank you for your suggestion. We changed the title
- Clearly this is not a systematic review nor a meta-analysis, it is thus unclear why the authors used the PRISMA guidelines. This is a non-sense. Certainly as those guidelines are not exploited further in the manuscript. For example, the quality of the studies is not reported, which is not a problem to me but this is recommended for a systematic review. There is thus inconstancy between the reported methodology and the structure and content of the manuscript.
You are right. We have modified this aspect in the new version
- The objective of the work is to summarize the scientific knowledge around hydration, nutrition, and metabolism at high altitude and to put them into the practical context of extreme altitude alpinism (>8000 m). I don’t understand why the authors specify the altitude here as the articles have not been selected on that parameter. Many articles deal with lower altitudes. I would suggest not to specify the altitude in the objective of the work.
Thank you, we have followed this recommendation.
- lines 11-125: “Anyway, it is difficult to predict the magnitude of the differences in climbing or 122 walking performance at varying altitude, as there are other factors as motivation, technical 123 abilities, biomechanical efficiency, experience or the capacity of tolerating mental stress 124 and thermal discomfort that cannot be measured (7,11).” This statement is not correct as each variable mentioned here can be quantified by physiological tests and/or questionnaires.
We changed this too
- Maybe my most fundamental comment is the confusion between the original/novel input (nutrition/hydration) and the already known (physiology). Everything dealing with physiological adaptations is already well described in the literature compared to nutrition and hydration. To me, the title reflects the novelty as it only mentions nutrition and hydration. Then, the objective is probably a bit less novel as it proposes to summarize the knowledge around metabolism, which is already treated in the literature. I would suggest to move everything that deals with physiology and performance (lines 68 to 262) to the introduction as it is not specific to the new input to the literature and does not respond to the key words used for the search strategy.
Thank you for this valuable point of view. We have modified the structure of the article as suggested.
Reviewer 2 Report (New Reviewer)
ijerph-2141975
Nutrition and hydration for high altitude alpinism: A practical approach
Summary:
ijerph-2141975 “Nutrition and hydration for high altitude alpinism: A practical approach” is a literature (non-statistical) review of the dietary needs during physical activity at extreme altitude. However, it tends to be more of a review of high altitude environmental physiology than of the dietary needs at altitude.
Overall comments:
Some editing for English language use and word choice is recommended, i.e. at the end of line 39 on page 1, it should be “decision making” instead of “decision taking.”
References could be cited more frequently throughout. For example, it is the third paragraph of the Discussion before the second reference is cited.
The manuscript is about the dietary needs at extreme altitude; however, much of the initial discussion describes environmental physiology rather than the results of the literature search. This seems better suited to the introduction. More discussion of nutrition would be useful and the recommendations do not appear to be evidence-based (this could be due to the limited citation of references).
Specific comments
Page 1, Line 44: Word choice. “This report pretends . . .” Pretends means “speak and act so as to make it appear that something is the case when in fact it is not”
Figures and Tables:
The later tables are rather long . . .
References
No comments.
Author Response
Overall comments:
Some editing for English language use and word choice is recommended, i.e. at the end of line 39 on page 1, it should be “decision making” instead of “decision taking.”
We changed that, thank you
References could be cited more frequently throughout. For example, it is the third paragraph of the Discussion before the second reference is cited.
We have changed the structure of the original text and added more references.
The manuscript is about the dietary needs at extreme altitude; however, much of the initial discussion describes environmental physiology rather than the results of the literature search. This seems better suited to the introduction. More discussion of nutrition would be useful and the recommendations do not appear to be evidence-based (this could be due to the limited citation of references).
As we changed the structure of the manuscript, we think that the re-arrangement of several parts of the text is in agreement with your suggestion
Specific comments
Page 1, Line 44: Word choice. “This report pretends . . .” Pretends means “speak and act so as to make it appear that something is the case when in fact it is not”
Thank you for that comment. We changed that
Figures and Tables:
The later tables are rather long . . .
You are right, but we tried to summarize as much as possible the information.
Reviewer 3 Report (New Reviewer)
In the abstract, the aim of this study should be stated first. Afterwards, summary results should be given about the reviewed articles. Then suggestions can be given.
Author Response
In the abstract, the aim of this study should be stated first. Afterwards, summary results should be given about the reviewed articles. Then suggestions can be given.
Thank you. We have changed the abstract according to your comment.
This manuscript is a resubmission of an earlier submission. The following is a list of the peer review reports and author responses from that submission.
Round 1
Reviewer 1 Report
The revision is largely improved and the concerns are mostly addressed
Author Response
Thank you for your constructive assessment and kind comments
Reviewer 2 Report
The article continues to have serious flaws.
1. If it is a systematic review, this should be specified in the title of the article.
2. The abstract is too long, it should be more to the point and summarize the recommendations made. This comment was made previously, and was not corrected by the authors.
3. The introduction is very poor, there is only one bibliographic citation. It is essential to justify why the systematic review was carried out. This comment was also made in the first review.
4. What was the PICO strategy to perform the search?
5. What was the search equation? This comment was made in the first review. It is necessary to know the screening of the articles. It is also necessary to know if any were added by snowball system.
6. It is necessary to make a flow chart of the search used as indicated in the PRISMA guidelines.
7. Why was the quality of the articles not evaluated using scales such as the PEDro?
8. The methodology part is very poor.
9. There is no results section. This is important to present the results found after the search.
10. The authors confuse the results section with the discussion section.
Author Response
Thank you for your constructive assessment
This is not a systematic review. As indicated in the title, this document's main aim is to provide some suggestions, based on the little scientific evidence available, for proper nutrition and hydration for climbers at extreme altitudes. We have replaced the word “review” with “report” in the only case where we have inadvertently used this term (second paragraph of the Introduction section).
Our primary approach in this document is based on the experience and knowledge gained by a licensed mountain guide with more than thirty years of professional experience, a sports and mountain medicine physician who served on the Medical Committee of UIAA (International Climbing and Mountaineering Federation) for about five years on behalf of Spain, and a physiologist with extensive experience in the study of exercise physiology and hypoxia.
According to the recommendations of the reviewer, we performed an extensive bibliographic review. As we declare in our document, relevant literature regarding nutrition and high-altitude expeditions is scarce. Regrettably, and as we expected, our search was not able to add more references to those we had previously collected and known.
In addition, is important to remark that the information on nutrition and hydration in high-altitude expeditions does not fit to any standard "experimental design", and therefore it is not reasonable to apply a quality control system, such as PEDro, to the scientific evidence collected.
We would like to highlight that the rules referred to by the reviewer for applying to systematic reviews have been devised to bridge discrepancies between studies with multiple, sometimes contradictory, and often confusing evidence. However, in our case, applying this strict mode of action would lead us to reject most of the available bibliography and discard all evidence.
We want to apologize if we have not been able to convey all that in our second version.
Finally, we wish to express that we believe that our document can be a relevant contribution in its current state of knowledge on this topic, since, despite not conforming to the rules of a systematic review, it provides practical information and recommendations of interest for the practice of extreme mountaineering.